# Quantifying patient preferences for symptomatic breast clinic referral: a decision analysis study

Aisling Quinlan,[1] Kirsty K O'Brien,[1] Rose Galvin,[1,2] Colin Hardy,[1] Ronan McDonnell,[1] Doireann Joyce,[3] Ronald D McDowell,[1] Emma Aherne,[1] Claire Keogh,[1,4] Katriona O'Sullivan,[1] Tom Fahey[1]

[1]HRB Centre for Primary Care Research, Department of General Practice, Royal College of Surgeons in Ireland, Dublin, Ireland
[2]Department of Clinical Therapies, University of Limerick, Limerick, Ireland
[3]Department of Surgery, National University of Ireland Galway, Galway, Ireland
[4]School of Computer Science and Statistics, Trinity College Dublin, Dublin, Ireland

**Correspondence to**
Professor Tom Fahey;
tomfahey@rcsi.ie

## ABSTRACT

**Objectives** Decision analysis study that incorporates patient preferences and probability estimates to investigate the impact of women's preferences for referral or an alternative strategy of watchful waiting if faced with symptoms that could be due to breast cancer.

**Setting** Community-based study.

**Participants** Asymptomatic women aged 30–60 years.

**Interventions** Participants were presented with 11 health scenarios that represent the possible consequences of symptomatic breast problems. Participants were asked the risk of death that they were willing to take in order to avoid the health scenario using the standard gamble utility method. This process was repeated for all 11 health scenarios. Formal decision analysis for the preferred individual decision was then estimated for each participant.

**Primary outcome measure** The preferred diagnostic strategy was either watchful waiting or referral to a breast clinic. Sensitivity analysis was used to examine how each varied according to changes in the probabilities of the health scenarios.

**Results** A total of 35 participants completed the interviews, with a median age 41 years (IQR 35–47 years). The majority of the study sample was employed (n=32, 91.4%), with a third-level (university) education (n=32, 91.4%) and with knowledge of someone with breast cancer (n=30, 85.7%). When individual preferences were accounted for, 25 (71.4%) patients preferred watchful waiting to referral for triple assessment as their preferred initial diagnostic strategy. Sensitivity analysis shows that referral for triple assessment becomes the dominant strategy at the upper probability estimate (18%) of breast cancer in the community.

**Conclusions** Watchful waiting is an acceptable strategy for most women who present to their general practitioner (GP) with breast symptoms. These findings suggest that current referral guidelines should take more explicit account of women's preferences in relation to their GPs initial management strategy.

## Strengths and limitations of this study

► Strength of this study is that it accounts for women's preferences in relation to a watchful waiting or immediate referral strategy when faced with symptoms that could be due to breast cancer.
► Weakness is that the preferences were elicited in asymptomatic women. Women with symptoms may express different preferences.
► The impact of eliciting preferences should be done in the context of a randomised trial of a decision aid that uses decision analysis as a form of shared decision-making for women with symptoms that could be due to breast cancer.
► Taking account of women's preferences might mean that significantly fewer women would be referred to symptomatic breast clinics with a consequent saving in investigations and costs, and reducing the burden of 'medicalisation' in women with benign breast disease.

## INTRODUCTION

Between 2009 and 2011, it was reported that 31% of cancers in Irish women were due to breast cancer, making it the most prevalent cancer in Irish women.[1] In response, the National Cancer Control Programme clinical guidelines were introduced to streamline the referral process to symptomatic breast units. At present, general practitioners (GPs) act as gatekeepers for referral of women with breast symptoms. Referral to symptomatic breast clinics results in triple assessment: a three-step process comprising clinical, radiological and histological examination after fine-needle aspiration.[2] Over the past 10 years, the number of women referred to symptomatic breast clinics in Ireland has significantly increased, despite the fact that there has been no rise in breast cancer incidence.[2] In 2006, there were 23 575 new referrals with 2137 breast cancer cases diagnosed; in 2009, there were 32 249 referrals and 1879 breast cancer cases diagnosed; and in 2010, 37 631 new referrals with 2012 new breast cancer cases diagnosed.[3 4] Consequently, the malignant:benign ratio has altered from 1:10 in 2006, to 1:16 in 2009, to 1:18 in 2010.[4] The rise in these referrals for triple assessment

without an increase in breast cancer diagnosis means that more women are being exposed to invasive diagnostic investigations.

Clinical prediction rules (CPRs) are clinical tools that quantify the independent impact of factors from a patient's history, physical examination and diagnostic tests, and stratify patients according to the probability of having a target disorder.[5] Recently, CPRs have been derived and validated to calculate a patient's risk of breast cancer,[6 7] and have been proposed as methods to discriminate between patients at high risk of breast cancer from low-risk patients. Such an approach may serve to decrease the number of unnecessary referrals to symptomatic breast clinics in women with a low probability of breast cancer. Patients who are considered low risk could, as an alternative, undergo a watchful-waiting strategy (up to 6–8 weeks) with their GP. However, it is unclear from the current literature if patients who are considered low risk would find this acceptable.

Decision analysis is a technique to aid decision-making when uncertainty exists over the balance between benefits and risks of treatment.[8] Decision trees have been used to assess how patient's preferences impact on diagnosis and treatment for conditions such as ovarian cancer[9] and atrial fibrillation.[10] To date, little is known about how women who have symptomatic breast problems feel about the options of immediate referral or watching waiting in terms of referral to a symptomatic breast clinic. The aim of this study is to assess the impact of patient preferences for immediate referral to a specialist breast clinic or watchful waiting in primary care. We developed

a decision tree and elicited patient's preferences for these alternative diagnostic strategies by combining individual preferences with probability estimates of breast cancer using decision analysis.

## METHODOLOGY
### Study design
This study used decision analysis in the form of utility assessment and folding back a decision tree. The Strengthening the Reporting of Observational Studies in Epidemiology standardised reporting guidelines were followed to ensure the standardised conduct and reporting of the research.

### Developing the decision tree and health scenarios
Literature searches were conducted in medical databases such as PubMed, the Cochrane library and Web of Science to assist with the development of the decision tree and health scenarios. A specialist registrar with clinical experience in a symptomatic breast clinic was also consulted during this development phase. Following these consultations, the decision tree was constructed to portray the temporal course of the two diagnostic options (figure 1). The decision tree had 1 decision node with 2 diagnostic branches (a choice between watchful waiting and immediate referral), 9 probability or chance nodes, and 11 health outcome nodes. In accordance with the literature, the tree was designed such that only two branches emanated from each chance node.[11] The number of outcome nodes differed between the two

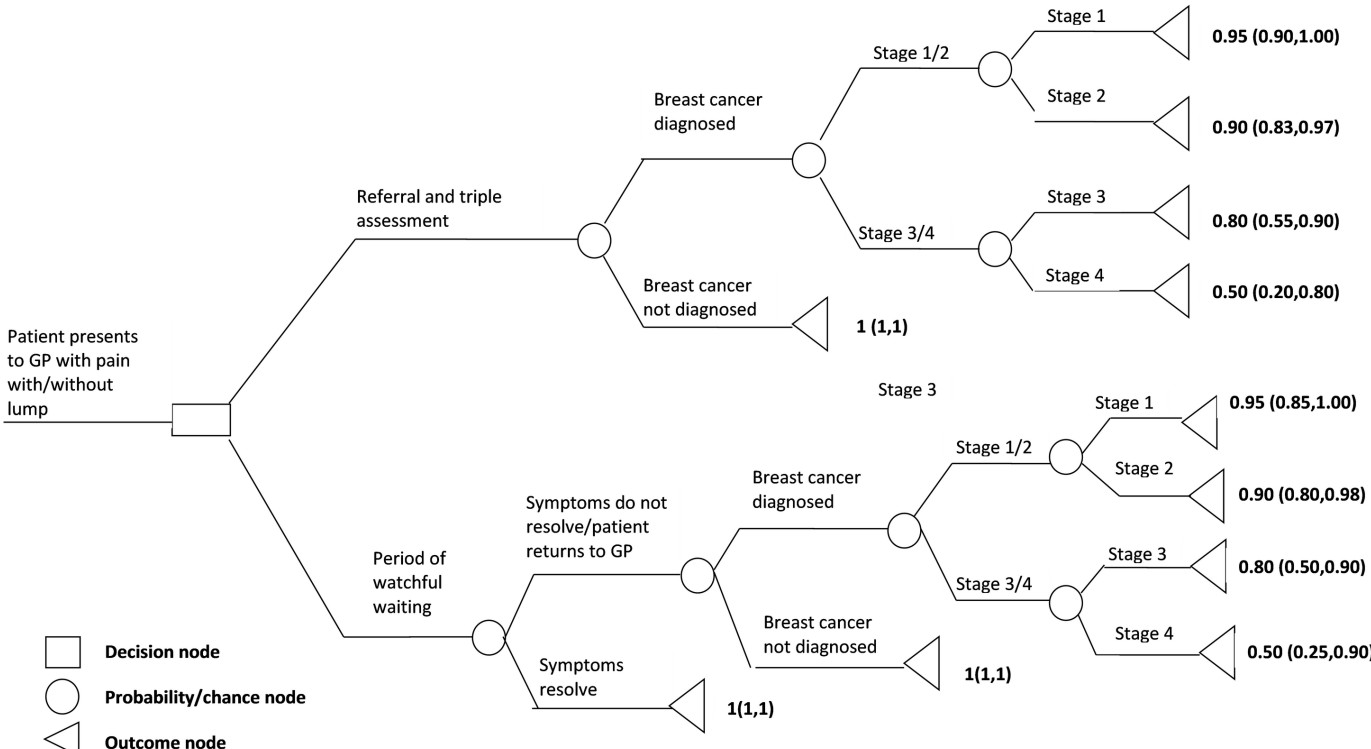

**Figure 1** Decision tree for health states in the treatment of breast cancer with utility values (median (IQR)) for each state. GP, general practitioner.

branches of the tree in order to accommodate the additional step (an observation period before breast cancer assessment among patients who return to their GP) in the watchful waiting branch of the tree. It is not always possible or appropriate for each branch of a decision tree to have the same number of outcome nodes as this will be dependent on the nature and temporal order of events in that branch.[12] The decision tree is 'balanced' as each branch has advantages and disadvantages, and hence it is not possible to readily identify a treatment path that is ideal in all respects.[11]

A total of 11 health scenarios were constructed; one for each outcome node. For this study, the standard gamble (SG) method of utility assessment was used.[13] [14] Briefly, participants were presented with a description of each health state, and its physical consequences and emotional consequences, and were given the opportunity to remain in the health state or to gamble it for perfect health, but with the risk of death if they took the gamble. Participants were presented with differing probabilities showing how the chance of perfect health and death varied with each other for each health state (the 'ping-pong method'). The utility value was one minus the risk of death they were willing to take in order to avoid the health state described. The 11 health states were presented in a random order. Details of the health states and the SG scenarios seen by the participants are provided in the online supplementary material.

Probability estimates for each chance node were obtained from the medical literature.[6] [7] [15–27] Where possible, relevant probability estimates for each chance node were combined using random effects meta-analysis (table 1). The same probabilities were used for each participant. It was not possible to estimate from the literature the probability of a diagnosis of breast cancer among patients who returned to their doctor following a period of watchful waiting. Clinical experience suggested this would be several percentage points higher in absolute terms than the probability of a diagnosis of breast cancer following referral and triple assessment (RTA), and an estimate of 10% was used.

**Table 1** Probability estimates used in the breast cancer decision tree analysis

| Treatment path | Outcome | Estimated probability (95% CI) | References |
|---|---|---|---|
| Referral and triple assessment (RTA) | Breast cancer diagnosed subsequent to RTA. | 0.06 (0.05 to 0.08) | 6 7 22 23 |
| | Stage 1 breast cancer among patients diagnosed with breast cancer following RTA. | 0.34 (0.26 to 0.42) | 15–21 |
| | Stage 2 breast cancer among patients diagnosed with breast cancer following RTA. | 0.46 (0.39 to 0.52) | 15–21 |
| | Stage 3 breast cancer among patients diagnosed with breast cancer following RTA. | 0.15 (0.13 to 0.17) | 15–21 |
| | Stage 4 breast cancer among patients diagnosed with breast cancer following RTA. | 0.05 (0.04 to 0.06) | 15–21 |
| Watchful waiting (WW) | Symptoms do not resolve/worsen and patient returns to general practitioner (GP) following a period of WW. | 0.34 (0.28 to 0.41) | 24 |
| | Breast cancer diagnosed among patients who returned to GP following WW. | 0.10 (0.05 to 0.18) | Clinical estimate |
| | Stage 1 breast cancer among patients diagnosed with breast cancer following WW. | 0.27 (0.20 to 0.33) | 25–27 |
| | Stage 2 breast cancer among patients diagnosed with breast cancer following WW. | 0.42 (0.36 to 0.47) | 25–27 |
| | Stage 3 breast cancer among patients diagnosed with breast cancer following WW. | 0.20 (0.16 to 0.22) | 25–27 |
| | Stage 4 breast cancer among patients diagnosed with breast cancer following WW. | 0.11 (0.09 to 0.13) | 25–27 |

## Participants and setting

Interviews were conducted over a 6-month period in Dublin from October 2014 to April 2015. Asymptomatic women aged 30–60 years were invited to participate in the study. Participants were recruited using university staff notice boards, social media and snowball sampling. The majority of interviews were carried out in a meeting room in the research building. Some interviews were conducted in peoples' place of work as this was more convenient for them. Written informed consent was obtained from all participants prior to participation. The interviews were conducted with a researcher present who was available to answer any questions and to obtain feedback about the interview. Prior to the interview, participants were given a baseline demographics questionnaire and a multiple-choice questionnaire to assess their knowledge of breast cancer. As this is a feasibility study for a proposed randomised controlled trial, no formal sample size was calculated. We excluded women with a current or previous history of breast cancer.

## Utility assessment

A computer-based assessment programme which presented and recorded participant responses was designed. This programme incorporated an introduction, an explanation of the interview process and presentation of each clinical scenario in random order using the SG method of utility assessment. An example of the SG was included for orientation, and definitions were provided for any medical terminology used. The purpose of using a computer programme was to allow the interviewer to limit his/her interaction with each participant, standardise utility assessment and minimise interviewer bias. A pilot study was carried out and modifications made to the computer programme and utility assessment process. See online supplementary table of health scenarios.

## Ethical approval and data protection

The data from the computer programme was stored with a participant ID only and saved to an encrypted password-protected desktop computer. To protect patient identity, a Microsoft Excel spreadsheet was created; containing just patient names and identification numbers, and this was stored in a secured, password-protected folder on the server and kept separate to the data from the interview.

## Decision analysis

The utility for each outcome, 'a number assigned to the quality of life a patient would attach to a particular outcome on a defined scale',[12] was calculated as 1-risk of death (expressed as a fraction) a participant was willing to take in order to avoid the health scenario as described. For each participant, an expected utility score for each diagnostic option was obtained by multiplying and summing the relevant probabilities and utilities. The branch which had the highest expected utility score was determined to be the preferred diagnostic strategy—watchful waiting or referral for triple assessment.

Sensitivity analysis was used to determine the robustness of the conclusion drawn from the decision tree.[28] For our study, each of the probability estimates derived from the literature was allowed to vary between the lower and upper limits of the 95% CI for that estimate, in order to assess the extent to which the preferred diagnostic strategy changed. The difference between the number of patients preferring watchful waiting using the upper limit of the 95% CI and the number of patients preferring watchful waiting using the lower limit of the 95% CI for each probability estimate (the swing) was squared and totalled; the impact of varying each of the probability estimates was measured as a percentage of the total squared-swing. The sensitivity analyses accounted for the fact that some probability estimates were used more than once by incorporating the conditioning event into the definition of the variables under consideration.[29] An additional two-way sensitivity analysis explored how women's preferred diagnostic option changed as the probability of diagnosis following watchful waiting varied relative to the probability of diagnosis following RTA.

Standard descriptive statistics were used to summarise the sample. Results were held to be significant if they referred to statistical significance on a two-sided test evaluated at the 5% level. All analyses were performed using STATA V.13 (StataCorp, 2013).

## RESULTS

A total of 36 women participated, with 35 interviews completed in full. The median age of participants was 41 years (IQR 35–47 years). The demographic details of participants are described in table 2. The majority of participants were employed (32, 91.4%), had a third-level (university) education (32, 91.4%) and knew someone with breast cancer (30, 85.7%). Less than half the study sample (15, 42.9%) reported having a prior mammography. There was borderline evidence of a negative correlation between participants' self-reported knowledge of breast cancer and their actual knowledge based on responses to a multiple-choice questionnaire (p=0.06).

When patient preferences were taken into account, 25 patients (71.4%, 95% CI 19% to 30%) preferred watchful waiting to referral as the initial diagnostic strategy. Sensitivity analysis showed that the overall preferred diagnostic strategy changed to referral when the probability of breast cancer following a period of watchful waiting increased; only 10 women favoured watchful waiting when the probability of breast cancer following a period of watchful waiting had increased to 18% (table 3). Swings in the group preference according to changes in the probabilities associated with each of the stage of breast cancer were small in comparison (table 3).

In terms of two-way sensitivity analysis, more than half the participating women still preferred watchful waiting when the probability of a diagnosis of breast cancer

**Table 2** Descriptive statistics for the study sample (n=35 participants)

| Variable | Median (IQR) | |
|---|---|---|
| Age (years) | 41 (35–47) | |
| Self-rated knowledge of breast cancer (maximum score 10) | 5 (4–7) | |
| Actual knowledge of breast cancer (maximum score 7) | 3 (2–4) | |
| | **Number of participants (%)** | |
| Location | | |
| Dublin and surrounds | 23 (65.7) | |
| West of Ireland | 10 (28.6) | |
| Overseas | 2 (5.7) | |
| Education | | |
| Secondary school | 3 (8.6) | |
| Third level (university) | 32 (91.4) | |
| Employed | | |
| Yes | 32 (91.4) | |
| No | 3 (8.6) | |
| Family carer | | |
| Yes | 18 (51.4) | |
| No | 17 (49.6) | |
| Know someone with breast cancer | | |
| Yes | 30 (85.7) | |
| No | 5 (14.3) | |
| Have had a mammography | | |
| Yes | 15 (42.9) | |
| No | 20 (57.1) | |

following a period of watchful waiting was 2.25 times as high as the probability of diagnosis following RTA (figure 2). This pattern remained consistent irrespective of the probability of breast cancer following RTA, estimated at levels of 4%, 6%, 8% and 10%.

## DISCUSSION
### Summary of main findings
This study shows that a strategy of watchful waiting in women with breast symptoms in primary care could be considered as an acceptable alternative to immediate referral to a symptomatic breast clinic, but this strategy of shared decision-making using decision analysis should be tested in a randomised controlled trial. Our observational findings suggest that women's own preferences should form an important element of the decision to refer to a symptomatic breast clinic and that these preferences should be accounted as part of a shared decision-making process. Sensitivity analysis shows that as the probability of breast cancer increases, the diagnostic strategy of referral

for triple assessment dominates. Two-way sensitivity analysis shows that, across the likely probability range of breast cancer under a watchful-waiting strategy, referral for triple assessment becomes the dominant strategy once the probability of breast cancer is two and a half times more likely than under a watchful-waiting strategy.

### Results in the context of the current literature
Although utility assessment has been used to evaluate the acceptability of various clinical states and treatment regimens in the breast cancer literature,[30–32] it has not been used to examine the acceptability of watchful waiting in low-risk patients presenting with signs and symptoms of possible breast cancer. A previous study that assessed the effects of alternative terms for ductal carcinoma in situ found that watchful waiting is acceptable to patients provided that research shows that watchful waiting is a safe and effective option.[33] The preliminary findings from our study suggest that women may be prepared to accept a conservative, non-invasive strategy of watchful waiting when the probability of breast cancer is low.

Watchful waiting has also been described as a key technique for more effective management of other cancers such as prostate cancer.[34] In prostate cancer, the use of prostate-specific antigen (PSA) testing has led to an increase in the incidence of prostate cancer but not a decrease in the mortality rate, suggesting that PSA testing often identifies low-risk cancers.[35] Given the serious and common side effects associated with treatment in prostate cancer, some studies have shown that some men prefer watchful waiting to treatment options.[36] In our study, the side effects of further diagnostic testing (triple assessment) are less onerous than treatment of localised prostate cancer, but many women would experience high levels of anxiety and stress on referral to a symptomatic breast clinic. This study suggests that many women would find it more acceptable to undergo watchful waiting for a period rather than participate in diagnostic testing when their risk of breast cancer is low.

A qualitative study examining women's views on overdiagnosis in breast cancer screening showed that 50% of the study population felt that they needed to make more careful personal decisions about screening and would consider watchful waiting as an alternative route if they were found to have a low-risk cancer.[37] Like our study, watchful waiting was found to be an acceptable strategy when the risk involved is low, and the health outcomes are positive.

### Strengths and limitations
This study has several strengths. We used an evidence-based approach, quantifying probabilities from the literature and developing health states that incorporate the common features of breast cancer. We tested and modified the wording of the health states and the computer programme in a pilot group of participants and refined our protocol based on feedback from these participants. We tried to limit any bias caused by the interviewer by

**Table 3** Sensitivity analysis for breast cancer patient treatment preference study

| Health scenario | Values for sensitivity analysis | | | Number of patients preferring WW to RTA | | | Swing | Swing$^2$ | %Swing$^2$ |
|---|---|---|---|---|---|---|---|---|---|
| | Lower limit for probability | Reference probability | Upper limit for probability | Lower limit for probability | Reference probability | Upper limit for probability | | | |
| Diagnosis of breast cancer following RTA | 0.05 | 0.06 | 0.08 | 23 | 25 | 27 | 4 | 16 | 3.21 |
| Stage 1 breast cancer among patients diagnosed following RTA | 0.26 | 0.34 | 0.42 | 26 | 25 | 21 | −5 | 25 | 5.02 |
| Stage 2 breast cancer among patients diagnosed following RTA | 0.39 | 0.46 | 0.52 | 26 | 25 | 24 | −2 | 4 | 0.80 |
| Stage 3 breast cancer among patients diagnosed following RTA | 0.13 | 0.15 | 0.17 | 25 | 25 | 25 | 0 | 0 | 0.00 |
| Stage 4 breast cancer among patients diagnosed following RTA | 0.04 | 0.05 | 0.06 | 25 | 25 | 25 | 0 | 0 | 0.00 |
| Symptoms do not resolve/worsen and patient returns to GP following period of WW | 0.28 | 0.34 | 0.41 | 27 | 25 | 24 | −3 | 9 | 1.81 |
| Diagnosis of breast cancer among patients returning to GP after period of WW | 0.05 | 0.10 | 0.18 | 31 | 25 | 10 | −21 | 441 | 88.55 |
| Stage 1 breast cancer among patients diagnosed following WW | 0.20 | 0.26 | 0.33 | 25 | 25 | 26 | 1 | 1 | 0.20 |
| Stage 2 breast cancer among patients diagnosed following WW | 0.36 | 0.41 | 0.47 | 25 | 25 | 26 | 1 | 1 | 0.20 |
| Stage 3 breast cancer among patients diagnosed following WW | 0.16 | 0.19 | 0.22 | 25 | 25 | 25 | 0 | 0 | 0 |
| Stage 4 breast cancer among patients diagnosed following WW | 0.09 | 0.11 | 0.13 | 26 | 25 | 25 | 1 | −1 | 0.20 |

Swing=number of patients preferring WW to RTA (upper limit for probability)–number of patients preferring WW to RTA (lower limit for probability).
Swing$^2$=swing×swing.
%Swing=swing$^2$×100/(total swing$^2$ (=498)).
GP, general practitioner; RTA, referral and triple assessment; WW, watchful waiting.

using a computer programme and providing the participant with written definitions and information. Examples of the SG were given to the participants prior to the interview, so participants had the opportunity to familiarise themselves with the method. Although the sample was relatively small, it included participants with a range of ages and knowledge of breast cancer. We acknowledge that the sample of participants is not representative of the population at large, that women were asymptomatic and that preferences may change or be different when faced

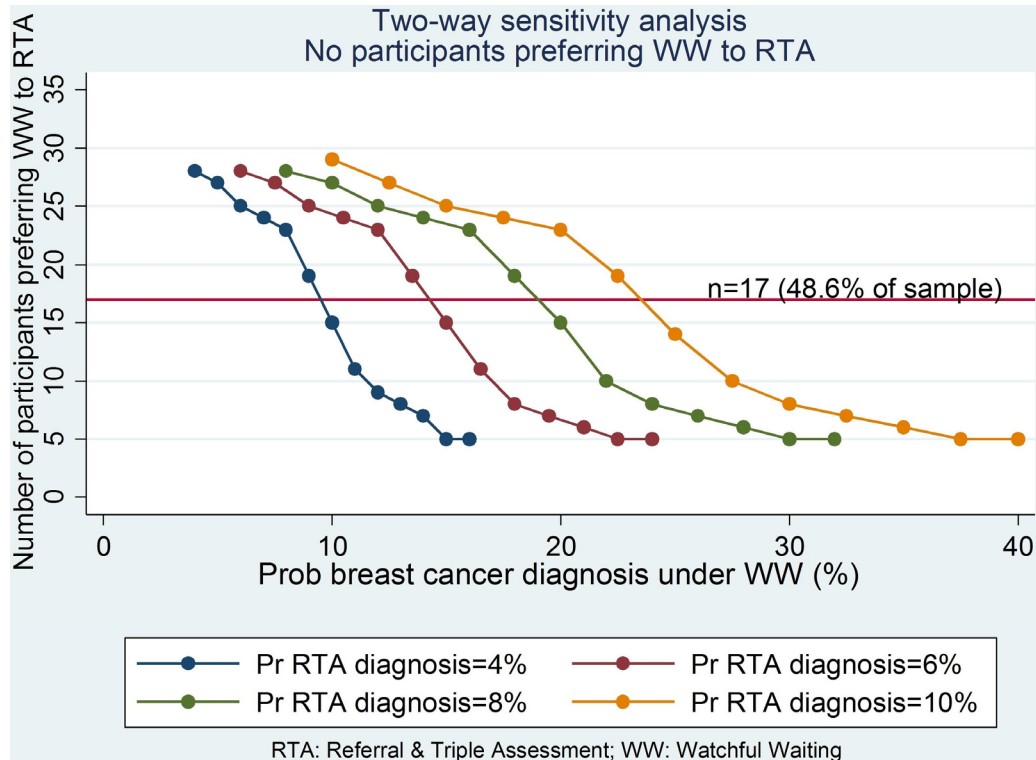

**Figure 2** Two-way sensitivity analysis according to probability under RTA (four levels) at incremental probability levels under WW. Pr/Prob, Probability.

with possible symptoms of breast cancer—a breast lump or nipple change. However, our results suggest that when preferences about a watchful-waiting strategy is elicited, many women are happy to adopt this approach. Based on these provisional results, we hope to develop a decision aid for women with symptomatic breast problems and test its value in a randomised controlled trial.

### Clinical and policy implications

The current referral process leads to many patients who are at low risk of breast cancer being referred to busy symptomatic breast clinics, with a risk of iatrogenic harm, increased patient anxiety and overmedicalising of a self-limiting condition. With more emphasis on the development of CPRs, it will be possible to identify and quantify each individual woman's risk of breast cancer. A recent study derived a CPR for predicting the risk of breast cancer for women presenting to their GP with breast symptoms.[7] A total of 6590 patients were included in the derivation study, and 4.9% were diagnosed with breast cancer. Independent clinical predictors for breast cancer were: increasing age by year (adjusted OR 1.08, 95% CI 1.07 to 1.09), presence of a lump (5.63, 95% CI 4.2 to 7.56), nipple change (2.77, 95% CI 1.68 to 4.58) and nipple discharge (2.09, 95% CI 1.1 to 3.97). Validation of the rule (n=911) also shows that the probability of breast cancer is higher with an increasing number of these independent clinical variables. By incorporating patient preferences with an individual's probability of breast cancer from a CPR, a strategy of watchful waiting is acceptable, cost-effective and likely to avoid considerable

iatrogenic harm. Clinical practice guidelines should incorporate probability estimates of breast cancer based on CPRs and urge elicitation of patient preference for alternative diagnostic options—watchful waiting or referral; more formally, decision aids that could be developed in conjunction with the symptomatic breast CPRs to enable the alternative options of watchful waiting or referral based on individual preferences.

### CONCLUSION

This study suggests that watchful waiting may be an acceptable diagnostic strategy for women who present to their GP with breast symptoms. These findings should be validated in a larger and more comprehensive study of patient preferences in women with breast symptoms in a randomised controlled trial. Clinical practice guidelines should reflect the importance of eliciting women's preferences for watchful waiting as an alternative to referral to a symptomatic breast clinic, particularly in women with low-risk clinical features.

**Contributors** All authors (AQ, KKOB, RG, CH, RMcD, DJ, RDMcD, EA, CK, KO'S and TF) were involved in the study conception and design. AQ, RG and KKOB acquired data for analysis. RMcD designed the computer-based utility analysis programme. CH and RDMcD performed statistical analysis. AQ interpreted the data and drafted the paper. KKOB, RG, RDMcD and TF critically revised the draft manuscript. All authors read and approved the final manuscript.

**Funding** Health Research Board of Ireland grant number HRC-2007-1.

**Competing interests** None declared.

**Patient consent** Obtained.

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
