## [Reviewer comments · BMJ Open]

ARTICLE DETAILS

TITLE (PROVISIONAL)	Quantifying patient preferences for symptomatic breast clinic referral: a decision analysis study
AUTHORS	Quinlan, Aisling; O'Brien, Kirsty; Galvin, Rose; Hardy, Colin; McDonnell, Ronan; Joyce, Doireann; McDowell, Ronald; Aherne, Emma; Keogh, Claire; O'Sullivan, Katriona; Fahey, Tom

VERSION 1 – REVIEW

REVIEWER	Jesse Jansen Sydney University, Australia
REVIEW RETURNED	23-Jun-2017

GENERAL COMMENTS	This study explores an important topic, the preference of asymptomatic women for watchful waiting vs testing after reading hypothetical scenarios about breast symptoms. The study used a standard gamble method. I am not an expert in this method but I feel that some basic information is missing in the methods section that makes it difficult to determine the overall quality of the design and interpret the results. - The sample size is 36 women, could the authors please justify that is this is suitable sample size for this type of studies? (it seems small)- Little detail is provided on what literature informed the decision tree and the factors included in the scenarios. This is an essential step and the quality of the SG task will depend on the quality of the information in the decision tree and health scenarios:o Decision tree: ideally a Table would be provided with the different nodes including a justification for its inclusion and the probability used (including references to relevant literature). In particular the probability of a diagnosis of breast cancer among patients who returned to their doctor following a period of watchful waiting should be motivated better, why 10%?o Health scenarios: how was the literature searched? Which search terms were used? Did the authors look at qualitative studies as well as quantitative information? Which factors did the authors identify and were there any factors that they decided not to include in the scenarios? Are they confident that these are factors important to women making these types of decisions?- Participants: women were asked if they knew anyone with breast cancer, did you also ask if they had any personal experience with breast cancer themselves or a family history? Did you measure perceived BC risk or BC worry? How was breast cancer knowledge measured?- As the authors explain in the introduction, watchful waiting is an approach that can potentially be used in low risk women (so in
--

	conjunction with risk stratification). Did the scenarios explain this to women? i.e. that they were estimated to be at low risk of BC? (I could not find it anywhere). Related to this, the authors acknowledge the importance of labelling/language in this context (e.g. McCaffery paper). How was watchful waiting defined/explained to women in the scenarios? Minor comments  - How do the Irish numbers mentioned in the introduction compare to international figures? - Literature: more recent literature on the terminology used around watchful waiting, active surveillance/active monitoring could be added. - Methods/participants: one of the recruitment methods was through staff notice boards- what type of companies? Health Industry or other? Which social media were used? - Methods: it is a computerized survey so would avoid the term interview as it is a bit confusing - Make sure abbreviations are written out in full at least once - Discussion p10 line 35: McCafferty should be McCaffery
--	--

REVIEWER	Dr Ray Moynihan Bond University, Centre for Research in Evidence- Based Practice
REVIEW RETURNED	27-Jun-2017

GENERAL COMMENTS	Thanks for opportunity to review this interesting and important study. While I am very sympathetic to this work, and to the expressed views/conclusions of the authors about the need to raise the profile of the watchful waiting option, I think the paper needs some important revision before re-consideration. A small problem is the lack of clarity in the abstract. A bigger issue is the need for clarity on whether this is a qualitative study – using interviews with a small group of women - or a quantitative “decision analysis” study – or a mix of both. If possible, it would be great for someone with expertise in this method to review the paper. And perhaps more clarity on the method- why it was chosen, why/how the sample size was selected, which STROBE guideline was used, etc – will help make things clearer. An important caveat to my review is that I am not a medical doctor, and I have no formal statistics training, so am unable to comment on the appropriateness or accuracy of the stats used. Having said all that, I strongly encourage the authors to revise and re-submit, as I think this is very valuable research on an extremely important health challenge. I have a list of specific comments below. Title : Page 1, Line 3. Title could be clearer. “symptomatic breast clinic referral” seems an uncommon phrase and I don’t immediately understand it. While it becomes clear later – you want the title to be as clear and compelling as possible. Page 2. The abstract is not as clear as it could be – I was confused about what the health scenarios might entail- and confused generally in the abstract – despite having a strong interest in this area. I suggest major re-write of the abstract – to make it very very clear. Page 2, Line 35. I don’t understand what is meant by “when individual preferences were accounted for”- and I don’t understand by what is meant by “triple assessment” Page 5, Line 23. I am not certain that the STROBE guideline is the most relevant here – and I wonder whether the COREQ checklist may be more appropriate- as a guide for how to report this- if it is
--

deemed a qualitative study. At the very least, I think, you need to be clearer about which STROBE guideline/checklist you were following- and ensure it was an appropriate one for your method.

Page 6, Paragraph starting at Line 11 : Would be good to be even clearer about the Standard Gamble methods – perhaps with examples of text used in this study – to help make it clear what exactly participants saw/responded to. (I noticed later during reading that text appeared at the end of the manuscript – but I think I am right in saying there was no mention of this material in the text of the manuscript—ie no link to Supporting Information file etc – forgive me if I missed that reference.)

Page 6: Recruitment: “Participants were recruited using staff notice boards, social media and snowball sampling” – this seems like a very important limitation - as there is no sense of how many recruited via which method—or how representative the sample was. I think it important to extend limitations section in Discussion – near page 11, line 40 + Page 7, Line 23. Again- need a link to the Supporting Information- and/or a box in the main manuscript – with an example of what people saw- to help reader understand.

Page 7, Line 45. You have stated info about ethics twice.

Page 7, Lines 50-54. This is critically important information- but as currently written- is not clear enough to readers unfamiliar with this method.

Page 8, Lines 11-35. As a non-biostatistician, I find this impossible to understand. I appreciate statistical methods are often not clear in written articles, but I think more needs to be done here to make this as clear as possible - to justify why this approach/method was chosen, and where possible, to cite more references to the literature (eg biostats literature, or decision analysis literature)

Page 9. Results.

I am still uncertain about whether this is best described as a qualitative study – given it involves interviews with 35 women. If it is a qualitative study – then the write-up would need to be done differently (using something like COREQ)- and I am not sure you would need so many numbers in the Results. If it is not a qualitative study – then you would likely need some explanation/justification for the sample size (perhaps its here, and I missed it) and the small number of participants would be a very important limitation.

Page 9 – last 2 paragraphs. Given how important these findings are – I think you need to make the reporting of these results as clear as possible. Again, despite my strong interest in this field, I am struggling to understand exactly what you are finding here. (Same goes for Page 10, lines 18-23)

Page 10- Line 38. The McCaffery et al study – cited as reference 35 (with me as a co-author) – was not with women with DCIS- but was a community sample. (“In a randomised comparison of terms for DCIS among a national community sample of women, interest in watchful waiting was high irrespective of the terminology used.” “The study was limited by its hypothetical design as women facing a real diagnosis of DCIS may respond differently to participants in our survey”).While that study’s findings of a high degree of openness to WW are important, I would be more cautious in conclusion drawn from this study in the following sentence in your manuscript. (In light of this comment - and with respect - I'd recommend double checking all the claims in your manuscript arising from a reference, to ensure that you are correctly citing the findings of other studies.)

Page 11- Line 40. You need to be much clearer about the limitations of this study – its much more convincing to read a paper where authors are not shy about their study limitations- but rather are very

	clear. Page 12, Line 14 – I would change “is” acceptable to “may be” Page 12, Line 33. The first line of the Conclusion- is I think, overstating your findings. This study was with “asymptomatic” women – yet you seem to be jumping to say these results apply to women with symptoms.
--	---

REVIEWER	John Benson Cambridge Breast Unit, Addenbrooke's Hospital, Cambridge CB2 0QQ
REVIEW RETURNED	14-Jul-2017

GENERAL COMMENTS	Patients present to breast clinics with a range of symptoms but two-thirds will have either a lump or an area of lumpiness (nodularity). Many breast complaints such as pain, lumpiness and nipple discharge may be sequelae of normal physiological changes taking place at different stages of reproductive life and in response to cyclical hormonal changes. Benign conditions such as fibroadenosis and fibroadenomas may present with lumps that fluctuate with the menstrual cycle or have decreased in size since first noticed by the patient. They are also more likely to be associated with pain and tenderness than a malignant breast lesion. A previous history of benign breast conditions, particularly cysts or fibrocystic change can be important indicators of the likely nature of any current breast problem. A proportion of those complaining of a breast lump will not actually have a discrete or dominant mass, but rather an area of focal nodularity which represents a prominent area of normal glandular tissue or an abnormality of normal development and involution (ANDI). The latter is a spectrum of changes in the breast parenchyma, epithelial elements and stroma. Distinction may be evident on clinical examination, but often further evaluation with ultrasound is necessary to confirm the presence of a discrete mass. Evaluation of breast referrals is based on the principle of “triple assessment” involving a combination of clinical examination, imaging and tissue biopsy. This yields an overall accuracy of almost 100% and permits the majority of patients attending ‘one-stop’ clinics to be reassured and discharged without further follow up. There is much overlap in the clinical features of benign and malignant breast conditions and physical examination alone has limited accuracy. This underscores the principle of triple assessment in which clinical examination is complemented by radiological imaging with or without some form of tissue biopsy. A review of almost 7000 patients undergoing ‘triple assessment’ within the Cambridge Breast Unit revealed an accuracy of 99.6% with only 0.4% of cases representing a ‘missed diagnosis’ on retrospective review. Although triple assess is accurate in terms of cancer detection, many women undergo invasive diagnostic investigations for non-malignant conditions or even for a B1 diagnosis (normal breast tissue). Over the past two decades the threshold for referring patients to breast clinics has fallen significantly with GPs referring a high proportion of patients presenting with the above spectrum of symptoms to one stop clinics which offer ‘triple assessment’. As pointed out by the authors, a mere 2,012 of 37,631 patients referred to breast clinics in 2010 were diagnosed with breast cancer. Most patients are now seen within 2 weeks and previous attempts to stratify patients in terms of risk for breast cancer have been unsuccessful with a significant number of cancers amongst patients
--

referred along non-urgent pathways. The proportion of patients referred to breast clinics with cancer was historically about 1 – 2 per 10 patients but now is much lower (1 in 14 – 18 patients). The authors have attempted to rationalize this referral process and seek the preferences of asymptomatic women (aged 30 – 60 years) for different diagnostic strategies – namely referral to a breast clinic (2 week wait) or watchful waiting. Clearly the acceptance of the latter will depend upon the perceived level of breast cancer risk and innate anxiety. The authors have employed previously validated tools (clinical prediction rules) to perform sensitivity analysis for various clinical scenarios associated with different probability estimates. Interestingly, when the upper probability estimate for breast cancer in the community is 18%, then patients prefer to be referred to a breast clinic and undergo triple assessment with clinical examination, radiological assessment and tissue biopsy if indicated. The study involved a relatively small number of participants (n = 35) who were young in terms of the age-specific incidence of breast cancer (mean = 41 years) and most were of moderately high educational attainment.

The use of clinical prediction rules in the context of breast disease might be questioned as there is much overlap in symptomatology, physical signs and indeed radiological indicators between benign and malignant disorders. Nonetheless, these have been validated for several diseases in the context of primary care and more recently have been developed for assessment of breast cancer risk. These tools have the potential for reducing referral to breast clinics and avoiding obligatory investigations (+/- biopsies) which are expensive and generate anxiety amongst patients.

It is quite reasonable for younger patients (<35 years) to be re-examined by GPs at a different stage in their menstrual cycle with a delayed referral after 6 – 8 weeks if symptoms persist. The current '2 week wait' does not allow for any spontaneous resolution/improvement of benign breast symptoms and cancellation of appointment.

The authors have combined individual patient preferences with probability estimates to generate a decision tree for management of patients according to two diagnostic options; patients are either referred initially to a breast clinic for triple assessment and some will be diagnosed with breast cancer (Stages 1 – IV) whilst the other group will undergo a period of watchful waiting with delayed referral for persistent symptoms. Using a total of eleven different scenarios from this decision tree (1 decision node, 9 probability nodes, 11 outcome nodes), utility values were calculated using the standard gamble method of utility assessment (utility value being 1 minus the risk of death patients were willing to take to avoid a particular health state). An arbitrary value of 10% was chosen as the excess probability of developing breast cancer from a delayed referral after a period of observation. This is a very difficult figure to estimate and is a potential weakness of this analysis. The highest expected utility score was used to determine the preferred diagnostic pathway for individual patients.

The authors found that a significant proportion of patients (n=25; 71.4%) preferred a watchful waiting policy but this fell when the probability of breast cancer after such waiting period increased to 18%. Interestingly, two-way sensitivity analysis showed that a watchful waiting policy was still acceptable when the probability of

	breast cancer was more than twice as high compared with immediate referral for triple assessment. This paper provides valuable data on behaviour of women in terms of low breast cancer risk; although it is often assumed that all women are anxious and many with benign symptoms believe they might have breast cancer, a high proportion of women seem prepared to accept watchful waiting for a limited period of time (when fully informed). It should be pointed out that managing low risk DCIS is slightly different to managing patients with breast symptoms which could potentially represent a more aggressive invasive cancer (perhaps with nodal involvement and micrometastatic disease). Those with low grade DCIS will have been fully assessed and more advanced stage disease excluded. The authors have listed strengths and limitations of this study and discussed policy implications. They emphasize risks associated with invasive investigations which include iatrogenic injury from biopsy but also anxiety generated from false positive results. Clinical prediction rules have identified established risk factors for breast cancer including age, presence of a lump, nipple change and nipple discharge (OR 2.09). Watchful waiting is unlikely to be recommended as a diagnostic option for older women presenting with a clinically discrete breast lump nor for those with (bloodstained) nipple discharge. The conclusions of this paper are justified – decision trees which incorporate patient preferences and information from clinical prediction rules can potentially reduce the numbers of referrals to breast clinics without incurring significant harm nor rise in breast cancer mortality secondary to delayed diagnosis of the disease. The paper is well-written in a clear and concise style with a comprehensive list of up-to-date references. The number of figures and tables is appropriate. I would recommend this paper for publication in BMJ online.
--	---

VERSION 1 – AUTHOR RESPONSE

Reviewer: Jesse Jansen

1. Sample size- this is a feasibility study for a potential randomized controlled trial of a decision aid. We have previously published decision aids using decision analysis to elicit individual patient preferences across conditions such as hypertension, menorrhagia and in pregnant women who are make a choice about their preferred mode of delivery.[1-3) We have added a sentence to the methods section to make clear that this is a feasibility study- “As this is a feasibility study for a proposed randomized controlled trial, no formal sample size was calculated”.
2. Literature on what informed the decision tree- extensive details of the literature and the probability estimates that inform the decision tree and health states are presented in Table 1.
3. Decision tree- Table 1 provides the information that the reviewer outlines. The most challenging literature to find was in relation to the probability of breast cancer in a primary care setting after an initial period of watchful waiting. We provide an estimate and a range of probability for this estimate based on prior probability of breast cancer in primary care and from clinical experience. We have acknowledged this uncertainty in the text of the methods . Please see paragraph “Probability estimates for each chance node were obtained from the medical literature [7,15-28]. Where possible, relevant probability estimates for each chance node were combined using random effects meta-

analysis (Table 1). The same probabilities were used for each participant. It was not possible to estimate from the literature the probability of a diagnosis of breast cancer among patients who returned to their doctor following a period of watchful waiting. Clinical experience suggested this would be several percentage points higher in absolute terms than the probability of a diagnosis of breast cancer following RTA and an estimate of 10% was used”.

4. Health scenarios- are generated from the decision tree which is based on our literature review and probability estimates outlined in Table 1, our own clinical experience and current clinical practice.

5. We did not ask the participating women any additional questions in relation to breast cancer, outside of what is presented in the paper. This is because our study concerns the use of decision analysis as a means of utility assessment and potential shared decision making. The additional questions the reviewer suggests are outside the remit of the paper.

6. Similarly, our paper was not concerned with framing issues in relation to risk or description of health states. We outline the health states and how they were presented to participants in the accompanying supplementary file.

7. International figures- it is beyond the scope of our paper to make international comparisons, but papers published in the BMJ show that the issue of over-referral and over-investigation in symptomatic breast clinics has also happened in the UK.

8. We feel that our terminology around watchful waiting is appropriate and up to date.

9. Recruitment was through a staff notice board principally, in addition to social media and snowballing of contacts. This is made clear in the methods section.

10. The method of utility assessment was via computerized utility assessment. It was not a computerized survey.

11. McCafferty has been corrected to McCaffery.

Reviewer: Ray Moynihan

1. This is a quantitative decision analysis study. No qualitative data is presented or included. We have amended the abstract as suggested by the reviewer to make this clearer: “: Decision analysis study that incorporates patient preference to investigate the impact of women’s preferences for referral or an alternative strategy of watchful waiting if faced with symptoms that could be due to breast cancer”.

2. Standardized reporting guideline- it is an observational study, but not of a formal epidemiological design- case control/cohort. We have included a STROBE statement as a supplementary file.

3. Abstract- has been re-written to make clearer that this is an observational study of individualized decision analysis.

4. Triple assessment is explained in the introduction- “: a three step process comprising of clinical, radiological and histological examination after fine needle aspiration”

5. This is not a qualitative study, so STROBE rather than COREQ seems more appropriate.

6. We have provided a link to the supplementary table re: standard gamble method.

7. Recruitment- we have outlined our recruitment strategy, and highlighted that this is a feasibility study. We have extended and outlined this in the limitations section as requested by the reviewer, by adding the following: “We acknowledge that the sample of participants is not representative of the population at large; that women were asymptomatic and that preferences may change or be different when faced with possible symptoms of breast cancer- a breast lump or nipple change. However, our results suggest that when preferences about a watchful waiting strategy is elicited, many women are happy to adopt this approach. Based on these provisional results, we hope to develop a decision aid for women with symptomatic breast problems and test it value in a randomized controlled trial”.

8. We have referenced supplementary material twice in the methods section – “See supplementary table of health scenarios”

9. We have removed the second mention of ethics committee approval.

10. Explanation of utility assessment and decision analysis- we appreciate that this is a difficult concept for the reader who is not familiar with the method. We have tried to explain both decision analysis and its impact on individual decisions as best as possible and in the same way that was done in our previous research.[1-3] Aside from have a formal utility assessment and decision analysis programme that is accessible via the web, we are not sure how we can explain the process in any more detail.

11. This is not a qualitative study- again, we have removed any comments that might suggest it is.

12. Apologies for our over-interpretation of the McCaffrey paper- we have amended to the following: “A previous study that assessed the effects of alternative terms for ductal carcinoma in-situ (DCIS). found that watchful waiting is acceptable to patients provided that research shows that watchful waiting is a safe and effective option [35]. The preliminary findings from our study suggest that women may be prepared to accept a conservative, non-invasive strategy of watchful waiting when the probability of breast cancer is low”.

13. Limitations- we have added additional limitations (outlined above)

14. Discussion and conclusion- we have moderated our discussion to “This study shows that a strategy of watching waiting in women with breast symptoms in primary care could be considered as an acceptable alternative to immediate referral to a symptomatic breast clinic, but this strategy of shared decision making using decision analysis should be tested in a randomized controlled trial” and our conclusion to “This study suggests that watchful waiting may be an acceptable diagnostic strategy for women who present to their general practitioner with breast symptoms. These findings should be validated in a larger and more comprehensive study of patient preferences in women with breast symptoms in a randomized controlled trial”

Reviewer: John Benson

1. We have incorporated additional comments about triple assessment definition and the increasing proportion of benign lumps seen in symptomatic breast clinics.

References

1. Montgomery A, Fahey T, Peters T. Decision analysis and information video plus leaflet for newly diagnosed hypertensive patients: a factorial randomised controlled trial. Br J Gen Pract 2003; 53: 446-453. PMID: 12939889.
2. Protheroe J, Bower P, Chew-Graham C, Peters T, Fahey T. Effectiveness of a Computerized Decision Aid in Primary Care on Decision Making and Quality of Life in Menorrhagia: Results of the MENTIP randomized controlled trial. Medical Decision Making 2007;27:575-584. PMID: 17898242.
3. Montgomery A, Emmet C, Fahey T, Patel R, Jones C, Ricketts I, Peters T, Murphy DJ. A randomised controlled trial of two decision aids for mode of delivery among women with a previous caesarean section. BMJ 2007;334:1305, doi:10.1136/bmj.39217.671019.55. PMID: 17540908.

VERSION 2 – REVIEW

REVIEWER	Phyllis Butow University of Sydney, Australia
REVIEW RETURNED	12-Jan-2018

GENERAL COMMENTS	This Irish study used a standard gamble methodology to assess women's preferences for immediate referral to a specialist breast clinic for triple testing, versus watchful waiting with their GP, following experience of breast symptoms that could be indicative of breast cancer.
--

	The study is very timely, given increasing numbers of referrals to specialist breast clinics in Ireland, without an increase in diagnoses. Methods allowed rigorous evaluation of how probabilities of a poor outcome influence women's preferences. The data would be useful for policy and practice. The authors suggest the findings support acceptability of a watchful waiting approach for women, and recommend a shared decision making approach incorporating women's preferences, formally elicited with a decision aid based on their methods. 1) However, the study sample is small (n=32); this was designed as a feasibility study for a proposed RCT of a decision aid. Given the well-educated sample, and the fact that participants did not currently have breast symptoms (acknowledged as a limitation), the findings may not be at all representative. 2) The standard gamble task was not sufficiently well described in the supplementary file for the reader to understand exactly what women were told, how the task was operationalised, and how probabilities were varied. It is not clear that participants were clearly told and understood that watchful waiting might delay diagnosis, and therefore increase their chances of having a later stage cancer when diagnosed. The chances of this may be very low, but it is certainly what women worry about. As the definitions provided to women are not shown, it is not clear what they understood triple testing to be comprised of. Were they told the chances of iatrogenic effects? Of false positives/negatives? In the different health states, women don't appear to have been told the implications of different stages of disease (except for stage 4, where they were told it could no longer be cured). 3) The recruitment process is not clear. On p.7, line 3, it says participants were recruited using staff notice boards and social media etc. What staff boards within which organisations? Was this a university sample? It does appear to be a very well educated sample. Were there any exclusion criteria? 4) In many places (Abstract, p.9, line 11 and in Table 2), education is recorded as second and third level. This is not terminology used in many countries. Could the authors clarify that this means school versus university? 5) P 7, line 15: The breast cancer knowledge questionnaire. Was this validated? Where did it come from? 6) There were multiple grammatical errors: a. P.4, line 21. I suggest, for consistency and ease of reading, rewriting as follows: "In 2006 there were 23,464 new referrals with 2137 cases diagnosed; in 2009 there were x new referrals and y cases diagnosed; in 2010 there were x new referrals and y cases diagnosed. b. P4, line 27: change malignant;benign to malignant:benign c. P.4, line 35: patients should be patient's
--	---

	d. P4, line 57: patient's should be patients' e. P. 5, line 5: watching waiting should be watchful waiting f. P. 5, line 7: rewrite to: the aim of this study is to assess the impact of breast cancer risk on patient preferences... g. P. 5, line 11, ...patients' preferences (add apostrophe) h. P.6, line 16: ...participants were presented (add were) i. P. 6, line 16: ...state, and its physical and emotional consequences... j. P. 10, line 45: beprepared should be be prepared k. P11, line 19: ...50% of the population study, should be 50% of the study population l. P.12, line 3 : test of value, should be test if its value m. P.12, line 32 ... increasing number of these independent clinical variables (s after variable) n. P. 42, lines 40-46. The whole sentence is not grammatical and needs to be reworded.
--	---

VERSION 2 – AUTHOR RESPONSE

Response to reviewer's comments:

Manuscript ID bmjopen-2017-017286 "Quantifying patient preferences for symptomatic breast clinic referral: a decision analysis study"

Thank you for the reviewer's comments and for giving us the opportunity to revise and clarify details from our study. Our response is detailed below in relation to the points raised by the reviewer. We address our comments mirroring the points raised by the reviewer:

- Overall comments- thank you for your initial positive comments on the overall aims and objectives and message of our paper that incorporating women's preferences is important when considering referral for breast symptoms in primary care.
- Feasibility- We accept that this is a small feasibility study. Our focus was on feasibility not generalizability and representativeness. This approach on initial feasibility,[1] and then testing validity and generalizability is an approach we have successfully adopted in the past in relation to decision aids for patients that use utility assessment as a formal way of quantifying and expressing patients' preferences.[2][3][4]
- Description of standard gamble- we do not feel that this this is an entirely fair comment. It is very difficult to demonstrate the standard gamble method, without real-time (computer) interaction. The slides that the patients went through on our computer programme show that we gave them an example of the standard gamble method (see slides 4 and 5 in the Appendix). Similarly during the whole standard gamble process, the participant was shown at the bottom of each slide the implications for their standard gamble in the context of each of the 11 health states that were assessed (please see all slides and the text under the heading "Standard Gamble"). As described in the methods section, we operationalized the utility assessment through a computer programme.
- Watchful waiting, definitions and health states- in terms of informing women about the consequences of watchful waiting and the definition of triple assessment, as we state in the methods section "The interviews were conducted with a researcher present, who was available to answer any questions." It is important to remember that watchful waiting or "test of time" is a standard diagnostic strategy in the context of clinical uncertainty used in general practice.[5] Unless every patient is referred immediately at initial consultation, then some form of watchful waiting is the only realistic clinical approach that a general practitioner in their gatekeeping role can adopt, even for "high stakes"

conditions such as breast cancer. Indeed, this is the underlying rationale for this study- to quantify and assess the impact of women's own preferences in the context of diagnostic uncertainty for a "high stakes" condition.

- Recruitment of participants- we have clarified recruitment and the sentence now reads: "Participants were recruited using university staff notice boards, social media and snowball sampling". There were no exclusion criteria aside from women with a personal history of breast cancer. We have added: "We excluded women with a current or past history of breast cancer."
- Education status- we have clarified that secondary level education means secondary school education and third level education means university. We have amended the abstract, text and table to make this clear.

- Breast cancer knowledge- this is not a validated questionnaire. We are not aware of any knowledge-based questionnaire for breast cancer.

- Typographical and grammatical errors:

- o We have re-written this sentence in the way suggested. It now reads: "In 2006, there were 23,575 new referrals with 2137 breast cancers cases diagnosed; in 2009 there were 32,249 referrals and 1879 breast cancer cases diagnosed; and in 2010, 37,631 new referrals with 2012 new breast cancer cases diagnosed [3][4]. Consequently the malignant:benign ratio has altered from 1:10 in 2006, to 1:16 in 2009 to 1:18 in 2010 [4]."

- o Benign:malignant is the terminology used in the national cancer reports that we cite.

- o All other errors have been corrected as suggested, aside from the aim of the study. The probability of breast cancer risk is independent of patient preferences. It is the combined utility and probability measure that determines an overall preference for different referral options. This is a fundamentally important point to make clear.

References

1. Protheroe J, Fahey T, Montgomery A, Peters T. The impact of patients' preferences in the treatment of atrial fibrillation: observational study of patient-based decision analysis. *BMJ* 2000; 320: 1380-1384.
2. Montgomery A, Fahey T, Peters T. Decision analysis and information video plus leaflet for newly diagnosed hypertensive patients: a factorial randomised controlled trial. *Br J Gen Pract* 2003; 53: 446-453.
3. Protheroe J, Bower P, Chew-Graham C, Peters T, Fahey T. Effectiveness of a Computerized Decision Aid in Primary Care on Decision Making and Quality of Life in Menorrhagia: Results of the MENTIP randomized controlled trial. *Medical Decision Making* 2007;27:575-584. Montgomery A, Emmet C, Fahey T, Patel R, Jones C, Ricketts I, Peters T, Murphy DJ. A randomised controlled trial of two decision aids for mode of delivery among women with a previous caesarean section. *BMJ* 2007;334:1305, doi:10.1136/bmj.39217.671019.55.
4. Heneghan C, Glasziou P, Balla J, Rose P, Thompson M, Lasserson D, et al. Diagnostic strategies used in primary care. *BMJ* 2009;338:b946.
5. Almond SC, Summerton N. Test of Time. *BMJ* 2009;338:b1878.